

# Broadband Radiative Quantities for the EarthCARE Mission: The ACM-COM and ACM-RT Products

Jason N. S. Cole[1], Howard W. Barker[2], Zhipeng Qu[1], Najda Villefranque[3], Mark W. Shephard[1]

[1] Environment and Climate Change Canada, Toronto, ON, Canada
[2] Environment and Climate Change Canada, Victoria, ON, Canada
[3] Laboratoire de Météorologie Dynamique, Paris, France

*Correspondence to*: Howard W. Barker (howard.barker@canada.ca)

**Abstract.** The EarthCARE satellite mission's objective is to retrieve profiles of aerosol and water cloud physical

properties from measurements made by its cloud-profiling radar, backscattering lidar, and passive multi-spectral

spectral imager (MSI). These retrievals, together with other geophysical properties, are input into broadband (BB)

radiative transfer (RT) models that predict radiances, and fluxes, commensurate with measurements made, and

inferred from, EarthCARE's BB radiometer (BBR). The scientific goal is that modelled and "observed" BB fluxes

differ, on average, by less than $\pm 10$ W m$^{-2}$. When sound synergistic retrievals from the ACM-CAP process are

available, they are acted on by the RT models. When they are not available, the RT models act on "composite"

atmospheric profiles of retrievals from individual sensors. "Compositing" is performed in the ACM-COM process as

described in this report.

The majority of this report describes the RT models, and their products, that make-up EarthCARE's ACM-RT

process. Shortwave (SW) and longwave (LW) flux and heating rate (HR) profiles are computed by 1D RT models for

each ~1 km nadir column of inferred properties. 3D RT models compute radiances for the BBR's three viewing

directions, with the SW model also computing flux and HR profiles; the 3D LW model produces upwelling flux at

just one level. All 3D RT products are averages over $5 \times 21$ km "assessment domains" that are constructed using

MSI data. A subset of ACM-RT's products is passed forward to the "radiative closure assessment" process that

quantifies, for each assessment domain, the likelihood that EarthCARE's goal has been achieved. As EarthCARE

represents the first mission to make "operational" use of 3D RT models, emphasis in this report is placed on differ-

ences between 1D and 3D RT results. For upwelling SW flux at 20 km altitude, 1D and 3D values can be expected to

differ by more than EarthCARE's scientific goal of $\pm 10$ W m$^{-2}$ at least 50% of the time.



## 1. Introduction

The EarthCARE satellite mission's primary objective is to make avant-garde observations of Earth's atmosphere that can be used to help improve representations of clouds and aerosols in numerical models that predict weather, air quality, and climatic change (Illingworth et al. 2015). Detailed descriptions of observations made by EarthCARE's cloud-profiling radar (CPR), backscattering lidar (ATLID), passive multi-spectral imager (MSI), and broadband radiometer (BBR), as well as the L2-retrieval algorithms that operate on them, are discussed in several papers of this special issue (Eisinger et al., 2022). EarthCARE's scientific goal is to retrieve cloud and aerosol properties with enough accuracy that when used to initialize broadband (BB) radiative transfer (RT) models to simulated top-of-atmosphere (TOA) BB fluxes for domains covering ~100 km$^2$, agree, more often than not, with their BBR-derived counterparts (Velázquez-Blázquez et al. 2022a) to within $\pm 10$ W m$^{-2}$ (ESA 2001). This comparison, which marks the end of the initial version of EarthCARE's formal "data production chain", provides a continuous radiative closure assessment of L2 retrievals with invaluable information to both L2-algorithm developers and data users.

The primary purpose of this paper is to describe and demonstrate the BB RT models used for both radiative closure assessment and provision of BB flux and heating rate (HR) profiles. Application of BB RT models to L2-retrieval products, along with auxiliary data, such as profiles of state variables and surface optical properties, will provide estimates of a range of diagnostic radiative flux and HR profiles. Examples of these products are presented here for ~6,200 km-long EarthCARE *test frames*, which are documented by Qu et al. (2022) and used throughout this special issue. Both 1D and 3D shortwave (SW) and longwave (LW) RT models are used. The 3D models produce TOA radiances; the 3D SW model also produces flux and HR profiles for all-sky conditions for a subset of ~100 km$^2$ assessment domains, while the 3D LW model produces upwelling fluxes at a single level. The number of assessment domains that can be processed per frame changes from frame-to-frame and will depend on computer resource availability during the mission as well as, to a lesser extent, cloud structure. Both SW and LW 1D models produce flux and HR profiles for each L2-column for all-sky, clear-sky (i.e., clouds removed), and pristine-sky (i.e., cloud and aerosol removed) conditions. This provides continuity with previous and ongoing missions such as CloudSat (Stephens et al. 2002) and CERES (Wielicki et al. 1996). All applications of RT models occur in the processor referred to as ACM-RT.

The current plan is for RT models to be applied to retrievals from the ACM-CAP process's CAPTIVATE algorithm

(Mason et al., 2022). ACM-CAP products, which are in the L2b class of products, are recognized formally as Earth-CARE's "best estimates" for they represent the most complete, synergistic, use of observations made by the CPR, ATLID, and MSI. Should ACM-CAP products not exist, the contingency plan is to use a *composite* back-up "best estimate" based on products that arise from retrieval algorithms that operate on measurements from a single active sensor. These products are in the L2a class. As such, the secondary purpose of this paper is to describe how the

composite cloud and aerosol profiles are generated within the ACM-COM process.

The following section provides an overview of the ACM-COM + ACM-RT processes and how they link with other processes. This is followed by a description of how EarthCARE retrievals are prepared for use in RT models. This includes presentation of the method used to create L2a-composite (back-up) cloud-aerosol profiles. In section 4 the SW and LW RT models are described along with atmospheric and surface optical properties. RT model results are

documented in section 5 making use of EarthCARE test frames. This includes showing the full extent of products from the 1D models and differences between SW and LW fluxes predicted by the 1D and 3D RT models. Section 6 provides a summary.

## 2. Overview of EarthCARE's radiation products

*Figure 1* encapsulates the main operations of ACM-COM and ACM-RT including its inputs and outputs. ACM-

COM prepares profiles of cloud and aerosol properties, produced by L2-retrieval processors (see Eisinger et al. (2022) for a summary), for use by the BB RT models in ACM-RT. Main operations of these processors are addressed in the subsequent two sections. The remainder of this section provides an overview of the components in *Figure 1*.

Arriving at ACM-COM are profiles of cloud and aerosol properties for each joint standard grid (JSG) column (Eisinger et al., 2022) along the L2-plane as retrieved by single active sensor L2a algorithms. ACM-COM also

receives similar profiles produced by the synergistic L2b CAPTIVATE algorithm in ACM-CAP, which utilizes ATLID, CPR, and MSI measurements (Mason et al., 2022). While studies to date suggest that ACM-CAP products will likely be EarthCARE's default "best estimates" (Mason et al., 2022a), this will not be known for sure until EarthCARE's "commissioning phase". Should ACM-CAP fail and only (some) L2a retrievals remain usable by RT





models, a contingency plan was developed in which L2a products are merged to form alternate "best estimate"

composite cloud-aerosol profiles. Compositing of L2a products is explained in section 3.2.

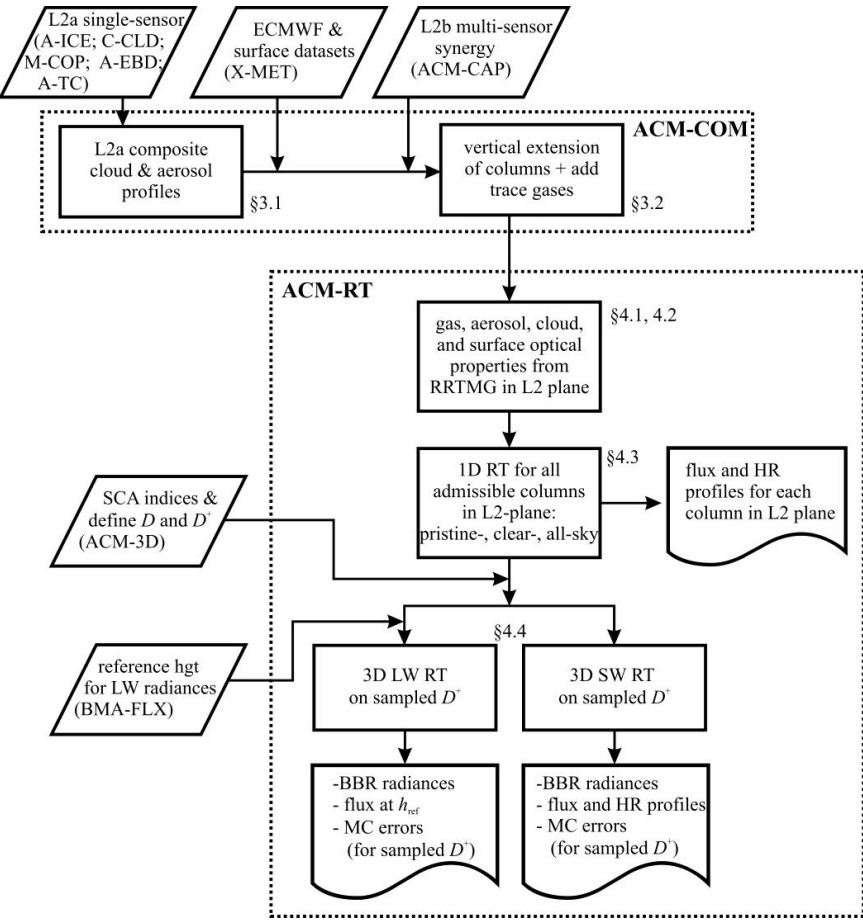

**Figure 1.** Flowchart summarizing the basic inputs to the ACM-COM and ACM-RT processes, their core operations, and their permanent output files. The operations are discussed in the sections that are listed next to them.

Regardless of whether ACM-CAP or alternate L2a-composite profiles are to be used by ACM-RT's RT models, they

need to be readied for use there. Hence, the last steps of ACM-COM take profiles of meteorological variables and

surface conditions, passed in respectively from the X-MET processor (Eisinger et al, 2022) and databases, and merge

them with ACM-CAP or L2a-composite products.



Following previous satellite missions (e.g., L'Ecuyer et al. 2008; Kato et al., 2013), ACM-RT computes SW and LW

BB flux and HR profiles by applying 1D RT models to each admissible JSG profile along the L2-plane. EarthCARE

makes a substantial step forward, however, with its operational use of 3D BB RT models for both SW and LW. For

consistency, 1D and 3D models use common descriptions of atmospheric and surface optical properties. Optical

properties for pristine atmospheres, free of aerosol and cloud, come from the Rapid Radiative Transfer Model for

General Circulation Models (RRTMG) (Iacono et al. 2008; Morcrette et al. 2008). RRTMG's SW and LW 1D two-

stream models compute flux and HR profiles for each JSG column along the L2-plane. The default is to use all

ACM-CAP profiles available in a frame. If no ACM-CAP profiles are available, or there is a request for its radiative

closure, we perform radiative transfer calculations for the L2a-composite profiles. These results get passed to

ACMB-DF (Barker et al. 2022) where they are averaged over "closure assessment domains" $D$ as dictated by ACM-

3D's scene construction algorithm indices (Qu et al. 2022).

The 3D RT solvers are Monte Carlo solutions of the plane-parallel 3D RT equation. They use the same gaseous,

aerosol, and cloud optical properties as the 1D models, but they also use detailed scattering phase functions. The SW

model produces profiles of fluxes and HRs, and TOA BB radiances commensurate with the BBR's three telescopes.

The LW model computes the same radiances, but its fluxes are only at a single "reference height" provided by the

BMA-FLX process (Velázquez-Blázquez et al., 2022a). All 3D RT computations are done for "radiation computation

domains" $D^+$ that consist of $D$ and buffer-zones around them (see *Figure 2*). Model-estimates of fluxes and radi-

ances, and any available uncertainties, are passed to the ACMB-DF processor (Barker et al. 2022) and averages over

$D^+$ are compared to BBR radiances and its model-derived fluxes (Velázquez-Blázquez et al., 2022a, Velázquez-

Blázquez et al., 2022b).



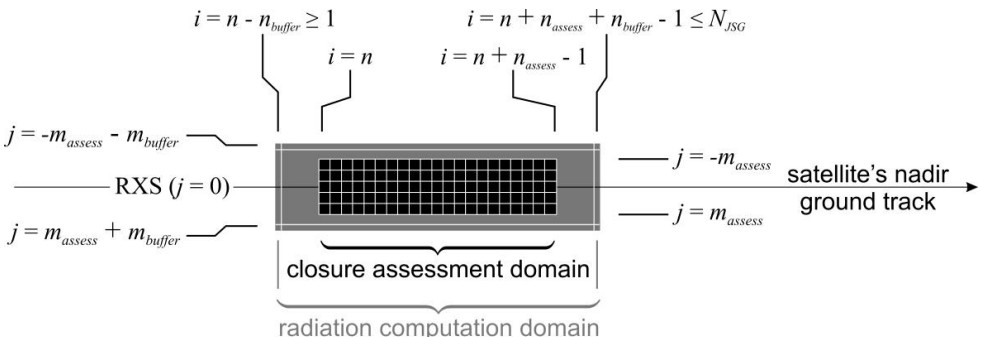

**Figure 2:** Schematic showing the radiative closure assessment domain $D$ (black) and the extended computation domain $D^+$ (shaded) that is the union of $D$ and its buffer-zones. These domains are centred on the L2a/L2b retrieved cross-section (RXS). See Qu et al. (2022) for details.

### 3. ACM-COM: Preparations for RT models and L2a-composites

As described in the following subsection, ACM-COM readies information from various L2-retrieval processes and X-MET for use by RT models in ACM-RT. This is followed by an explanation of how ACM-CAP's alternate *L2-composite profiles* are produced.

#### 3.1. Prepping L2-retrievals for RT models

The ACM-COM process begins by simply extracting, from X-MET files, information about atmospheric state as needed by all BB RT models. This includes profiles of pressure, temperature, humidity, and ozone concentration. Regarding aerosols, their classification information is provided by the AC-TC processor (Irbah et al. 2022) with extinction at 0.355 μm provided by A-EBD (Donovan et al. 2022). Six types of aerosols are considered: dust, sea salt, continental pollution, smoke, dusty smoke, and dusty mix. Those grid-cells in AC-TC that are classed as *cloudy*, *uncertain*, *missing*, or *noisy* are considered to be aerosol-free.

Additionally, ACM-COM adds the following minor molecular species to X-MET profiles: $CO_2$, $CH_4$, $N_2O$, CFC-11, CFC-12, CFC-22, and CCL4. These profiles from climatologies generated by J.-J. Morcrette and A. Bozzo (per. comm., R. Hogan 2013). Values are functions of month, pressure, and latitude.



### 3.2. Construction of "L2a-composite" cloud and aerosol profiles

In addition to ACM-CAP's synergistic retrievals, an alternate "best estimate" is produced, and can be used if ACM-

130    CAP data are unavailable, based on compositing L2a retrievals that use ATLID (A-ICE) (Donovan et al. 2022) and

CPR (C-CLD) (Mroz et al. 2022) observations.

The L2a-composite's cloud properties come from either A-ICE or C-CLD as subject to an indication of columnar

cloudiness from the M-COP processor (Hünerbein et al. 2022). If for a grid-cell A-ICE *or* C-CLD reports cloud

water content greater than zero, their cloud properties, as reported, enter into the L2a-composite. For grid-cells in

135    which both A-ICE *and* C-CLD report valid cloud properties with ice water content $IWC > 0$, aggregated normal-

ized uncertainties for $IWC$ and crystal effective radius $r_{eff}$ get computed, respectively, as

$$\sigma_{\text{A-ICE}} = \sqrt{\left(\frac{\sigma_{\text{IWC}}^{\text{A-ICE}}}{IWC^{\text{A-ICE}}}\right)^2 + \left(\frac{\sigma_{r_{eff}}^{\text{A-ICE}}}{r_{eff}^{\text{A-ICE}}}\right)^2} \quad \text{and} \quad \sigma_{\text{C-CLD}} = \sqrt{\left(\frac{\sigma_{\text{IWC}}^{\text{C-CLD}}}{IWC^{\text{C-CLD}}}\right)^2 + \left(\frac{\sigma_{r_{eff}}^{\text{C-CLD}}}{r_{eff}^{\text{C-CLD}}}\right)^2}, \qquad (1)$$

where $\sigma_{\text{IWC}}^{\text{A-ICE}}$, $\sigma_{\text{IWC}}^{\text{C-CLD}}$, $\sigma_{r_{eff}}^{\text{A-ICE}}$, and $\sigma_{r_{eff}}^{\text{C-CLD}}$ are processor-specific 1-sigma uncertainties. Ice cloud properties for

the product having $\min\left(\sigma_{\text{A-ICE}}, \sigma_{\text{C-CLD}}\right)$ enter into the L2a-composite. For grid-cells containing only liquid cloud,

140    C-CLD properties are used. Hence, L2a-composites resemble NASA's CloudSat-CALIPSO-CERES (C3M) product

(Kato et al. 2010).

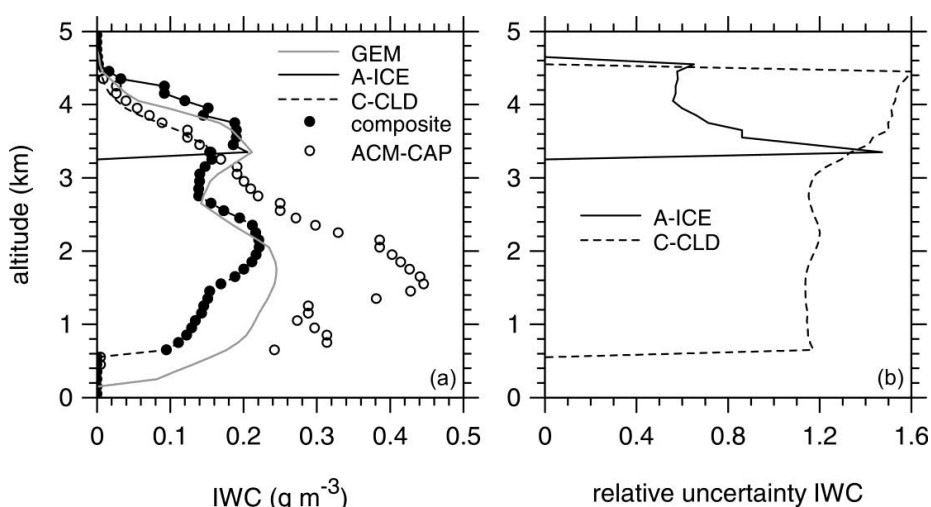

**Figure 3:** (a) Lines represent profiles of IWC directly from the test frame (simulated by GEM), as well as those retrieved by the L2a algorithms in processors A-ICE and C-CLD. Filled circles are layer values that ACM-COM's algorithm selected from A-ICE and C-CLD according to which one has the smallest aggregated relative uncertainty, defined by (1), as shown in (b). This profile, which has only ice cloud, is from the *Halifax* test frame at $63.67°$ N; $54.64°$ W.

*Figure 3* shows an example of this compositing process for a column from the *Halifax* frame (Qu et al. 2022). Only ice cloud was present, so both A-ICE and C-CLD reported hydrometeors. Above ~3.4 km ATLID's estimates have least uncertainty and so A-ICE values enter into the composite. At ~3.3 km the CPR value is least uncertain and so C-CLD's estimate is used. Below there ATLID failed to return a useable signal so with only CPR values remaining, they fill the remainder of ACM-COM's profile. The fact that in this example the "reference values" as simulated by the Global Environmental Multi-scale (GEM) model (Qu et al. 2022) match ACM-COM's better than do ACM-CAP's does not necessarily mean that ACM-COM's get used by the RT models. During the mission when ACM-CAP's exist, they get preference.

## 4. ACM-RT: Broadband radiative transfer models

As mentioned above, all of EarthCARE's RT models are based on RRTMG (Iacono et al. 2003, 2008; Morcrette et al. 2008). Like its computationally taxing progenitor (Mlawer et al. 1997; Mlawer and Clough 1998), RRTMG is built on the correlated *k*-distribution (CKD) method (Goody et al. 1989; Lacis and Oinas 1991). Broadband integrat-

ed flux and HR profiles are sums of calculations for 252 (140 LW + 112 SW) quadrature points spread over 30 spectral bands (16 LW + 14 SW). RRTMG is used widely in large-scale models (e.g., ECMWF, MPI, NCEP, NCAR, NASA/GSFC, LMD, CMA), and its verification has been documented elsewhere (e.g., Iacono et al. 2008; Oreopoulos et al. 2012). This section begins by describing atmospheric and surface optical properties, and follows with descriptions of the 1D and 3D transport solvers.

### 4.1. Optical properties: Atmospheric constituents

### 4.1.1. Gases

Molecular optical depths are computed by the CKD method in RRTMG_LW v4.85 and RRTMG_SW_v3.9 and used by both 1D and 3D RT models. Molecular absorption coefficients for RRTMG's *k*-distributions were obtained from the *line-by-line* RT model (LBLRTM), which has been evaluated against surface and laboratory observations (Clough et al. 2005; Shephard et al. 2009; Alvarado et al. 2012). LBLRTM's spectroscopic line parameters are essentially equivalent to HITRAN 2000 and HITRAN 1996 (SW) databases. Algorithmic accuracy of LBLRTM is 0.5% (Clough et al 2005) with limiting errors generally attributed to line shape and spectroscopic input parameters.

Wavenumbers for RRTMG's LW bands are: 10-350, 350-500, 500-630, 630-700, 700-820, 820-980, 980-1080, 1080-1180, 1180-1390, 1390-1480, 1480-1800, 1800-2080, 2080-2250, 2250-2380, 2380-2600, and 2600-3250 $cm^{-1}$. Molecular absorption optical depths are computed for: $H_2O$, $CO_2$, $O_3$, $N_2O$, $CH_4$, $O_2$, $N_2$, CFC11, CFC12, CFC22, and $CCl_4$. Additionally, CKD_v2.4's continuum model accounts for foreign- and self-broadening of lines for $H_2O$, $CO_2$, $O_2$, $O_3$, and Rayleigh scattering.

Wavenumbers for RRTMG's SW bands are: 2600-3250, 3250-4000, 4000-4650, 4650-5150, 5150-6150, 6150-7700, 7700-8050, 8050-12850, 12850-16000, 16000-22650, 22650-29000, 29000-38000, 38000-50000, and 820-2600 $cm^{-1}$, with the last band coded out of sequence for spectral continuity with the LW bands. Sources of extinction are: absorption by $H_2O$, $CO_2$, $O_3$, $CH_4$, $O_2$, $N_2$; and Rayleigh scattering.

For 1D SW RT, the Rayleigh scattering phase function is approximated as $p_{Ray}(\mu) = 1$, where $\mu = \cos\theta$ and $\theta$ is scattering angle. For 3D SW RT, on the other hand,


$$p_{Ray}(\mu) = \frac{3}{4}(1+\mu^2),$$
(2)

which is, as are all phase functions used here, normalized as

$$\frac{1}{2}\int_{-1}^{1} p_{Ray}(\mu)d\mu = 1.$$
(3)

Relative to LBLRTM, clear-sky RRTMG_LW BB fluxes at all levels are accurate to within $\pm 1.5$ W m$^{-2}$ ($\pm 1$ W m$^{-2}$

for direct-beam and $\pm 2$ W m$^{-2}$ for diffuse-beam), with HRs agreeing to within $\pm 0.2$ K day$^{-1}$ in the troposphere and

$\pm 0.4$ K day$^{-1}$ in the stratosphere. Likewise, RRTMG_SW's accuracies, at $\mu_0 \approx 0.7$, are within $\pm 3$ W m$^{-2}$ at all

levels, with HRs agreeing to within $\pm 0.1$ K day$^{-1}$ in the troposphere, and $\pm 0.35$ K day$^{-1}$ in the stratosphere.

### 4.1.2. Aerosols

As with gases, 1D and 3D RT models use the same spectral optical properties for aerosols: extinction coefficient

$\beta_{aero}$, single-scattering albedo $\omega_{aero}$ and asymmetry parameter $g_{aero}$. Spectral $\beta_{aero}$, $\omega_{aero}$ and $g_{aero}$ for the RT

models are averages over wavelength intervals listed above. To generate these in a manner consistent with the

retrievals, optics are computed following (Wandinger et al. 2022). The same radiative properties for basic aerosol

types are used, which are then externally mixed to generate radiative properties for the aerosol mixtures used in AC-

TC. Since profiles of aerosol extinction are provided at 355 nm, the ratio of $\beta_{aero}$ at each wavelength to $\beta_{aero}$ at 355

nm is used instead so the profiles of $\beta_{aero}$ computed from lidar data. Wavelength-resolved properties are then

averaged to each SW and LW spectral interval using the same weighting as for cloud radiative properties described

below.

In addition to these integrated optical properties, the 3D RT codes require scattering phase functions, which for

aerosols are represented by the Henyey-Greenstein (1941) function as

$$p_{HG}(\mu; g_{aero}) = \frac{1 - g_{aero}^2}{\left(1 + g_{aero}^2 - 2g_{aero}\mu\right)^{3/2}},$$
(4)



which satisfies

$$g_{aero} = \frac{1}{2} \int_{-1}^{1} p_{HG}\left(\mu; g_{aero}\right) \mu \, d\mu. \tag{5}$$

Owing to the size and irregularity of aerosol particles, and retrieval uncertainties, use of (4) is reasonable.

### 4.1.3. Liquid clouds

The standard version of RRTMG uses Hu and Stamnes's (1993) parametrizations for spectral $\beta$, $\omega_0$, and $g$ of

liquid droplets. For EarthCARE's 1D and 3D RT models, however, these have been replaced by more precise Lo-

renz-Mie calculations tabulated for ranges of droplet effective radii $r_{eff}$ and effective variances $v_{eff}$, which are

defined, respectively, as

$$r_{eff} = \frac{\int_{0}^{\infty} n(r) r^3 dr}{\int_{0}^{\infty} n(r) r^2 dr} = \frac{\langle r^3 \rangle}{\langle r^2 \rangle}, \tag{6}$$

and

$$v_{eff} = \frac{\int_{0}^{\infty} \left(r - r_{eff}\right)^2 n(r) r^2 dr}{r_{eff}^2 \int_{0}^{\infty} n(r) r^2 dr} = \frac{\langle r^2 \rangle \langle r^4 \rangle}{\langle r^3 \rangle^2} - 1, \tag{7}$$

where $r$ is droplet radius. Droplet size distribution $n(r)$ was assumed to be

$$n(r) = \frac{N}{\Gamma(v)} \left(\frac{v}{\langle r \rangle}\right)^{v} r^{v-1} \exp\left(-\frac{rv}{\langle r \rangle}\right), \tag{8}$$

where $\langle r \rangle = v r_{eff} v_{eff}$ and $v = \left(1 - 2 v_{eff}\right)/v_{eff}$.

Lorenz-Mie computations (Wiscombe 1980), using Segelstein's (1981) refractive indices, were performed for $r$

between 0.01 and 120 μm in increments of 0.05 μm, and for wavelengths $\lambda$ between 0.25 and 100 μm in increments

of: 0.02 for $0.25 < \lambda < 2$ μm; 0.04 for $2 < \lambda < 3$ μm; 0.05 for $3 < \lambda < 10$ μm; 0.07 for $10 < \lambda < 20$ μm; and 0.1 for 20

$< \lambda < 100$ μm. Phase functions and optical properties were integrated over RRTMG's spectral intervals for combina-

tions of $r_{eff}$ and $v_{eff}$ : $r_{eff}$ from 0.5 - 40 μm in increments of 0.5 μm; and $v_{eff}$ from 0.02 - 0.4 in increments of 0.02

μm. Spectral weightings for SW bands were downwelling irradiances averaged at the tropopause and surface from

LBL data (Iacono et al. 2008) for the tropical atmosphere at solar zenith angle $\theta_0 = 0°$. For LW bands, weightings

were the Planck function at 275 K. In the RT models, values of $r_{eff}$ and $v_{eff}$ get rounded to the nearest value in the

table, which usually results in errors for $\beta$, $\omega_0$, and $g$ of less than $\pm 1\%$.

As the 3D RT models are Monte Carlo solutions, they use normalized tabulated scattering phase functions $p(\mu)$

for droplets. Broadband, spectrally-integrated $p(\mu)$ have 1,800 equal angular bins, and their cumulative sums, as

functions of $\mu$, were computed by

$$R(\mu_s) = \frac{1}{2} \int_{\mu_s}^{1} p(\mu) d\mu, \tag{9}$$

where $\mu_s$ is cosine of scattering angle, with $R(\mu_s = 1) = 0$ (forescatter) and $R(\mu_s = -1) = 1$ (backscatter). For

efficiency, tables of $\mu_s$ were constructed for 1800 equally spaced values of $R$; when a scattering event occurs, a

uniform pseudo-random number gets generated $R \in [0,1]$, linear interpolation sets $\mu_s$, which is used to update a

photon's direction cosines.

### 4.1.4. Ice clouds

Values of $\beta$, $\omega_0$ $g$, and scattering phase functions for ice clouds as used in the 1D and 3D RT models are based on

Yang et al.'s (2013) theoretical functions for 11 crystal habits: droxtals, prolate spheroids, oblate spheroids, solid

columns, hollow columns, aggregates composed of 8 solid columns, hexagonal plates, small aggregates composed of





5 plates, large aggregates composed of 10 plates, solid bullet rosettes, and hollow bullet rosettes. Maximum dimension for each habit ranges from 2 μm to 10,000 μm for 189 discrete sizes. Three surface roughness conditions were considered for each ice habit: smooth, moderate, and severe. Each constituent has volume, projected area, effective size, extinction efficiency, $\omega_0$, and $g$. Their scattering phase functions are tabulated at 498 unequal angles, but were transformed into 1,800 equal angular bins for use in (9).

To make this dataset's size suitable for operational use, optical properties were averaged over $\lambda$ and assumed distributions of habit, size, and roughness that were derived from CALIPSO observations (Baum et al. 2011). Resulting phase functions and optical properties are functions of effective diameter which is defined as

$$d_{eff} = \frac{3}{2}\frac{\sum_i \int V_i(D)n(D)f_i(D)dD}{\sum_i \int A_i(D)n(D)f_i(D)dD}, \tag{10}$$

where $V$, $A$, and $D$ are geometric volume, orientation-averaged projected area, and maximum dimension of ice

particle, respectively. $n(D)$ denotes crystal size distribution, and $f_i$ indicates the percentage of each ice particle habit and roughness. Values of $d_{eff}$ range from 10 μm to 120 μm in increments of 5 μm. Band-averaged optical properties were computed using the same weightings as in (10) while also weighting for spectral irradiance and then integrating over RRTMG's spectral intervals. Spectral weight for SW bands was the TOA spectrum while for the LW it was the Planck function at 250 K (per. comm., B. Yi, 2013).

**4.1.5. Rain and snow**

Retrievals of ice cloud water included snow so the optics and radiative effect for snow are not computed explicitly. Rain water content is retrieved separately from non-precipitating liquid cloud. Its single scattering properties are defined in a manner similar to liquid cloud droplets (Section 4.1.3) and applicable to effective radii from 10 μm to 120 μm in increments of 5 μm.



### 4.2. Optical properties: Underlying surfaces

Both 1D and 3D RT models require surface spectral albedos and emissivities. Implicit in the 1D models is the assumption that reflection and emission are Lambertian. The SW model can handle different albedos for direct and diffuse irradiance.

The snow-free surface albedo over land for visible (0.3-0.7 μm) and infrared (0.7-5.0 μm) bands were calculated from climatological BRDF parameters for 16-day periods based on 12 years (2002-2013) of MODIS MCD43GF data (Schaaf et al. 2002). Terrestrial snow albedo data for the same spectral bands are based on Moody et al. (2007) whose calculations are based on five years (2000–2004) of climatological statistics of Northern Hemisphere white-sky albedos for 16 International Geosphere–Biosphere Program (IGBP) ecosystem classes when accompanied by the presence of snow on the ground. For ice-covered land or water surfaces, broadband average albedo over 16,000 - 50,000 cm$^{-1}$, as provided by X-MET (via ECMWF), will be used.

Ideally, the 3D RT models should include bidirectional reflection and emission functions; such as Rahman et al.'s (1993) land surface model, which is in EarthCARE's SW 3D RT code. For land surfaces, however, global parameters are lacking. Hence, spectral albedos and the Lambertian assumption are also used.

For open water surfaces, spectrally-independent ocean albedo is governed by

$$\alpha_p = 0.021 + x^2 \left( 0.0421 + x \left( 0.128 + x \left( -0.04 + x \left( \frac{3.12}{5.68 + w} \right) + \frac{0.074x}{1 + 3w} \right) \right) \right), \qquad (11)$$

where $x = 1 - \cos\theta_i$, $\theta_i$ is zenith angle of an incident photon, and $w$ is surface wind-speed (m s$^{-1}$) (Hansen et al. 1983). The 3D RT SW model uses Cox and Munk's (1956) ergodic wave model to describe the probability of a SW photon incident at the surface being reflected toward the BBR. As such, simulated radiances capture Sun-glint; the effects of which will be tempered by EarthCARE's orbit and MSI design (Illingworth et al. 2015). In addition, hemispheric infrared emissivities for each RRTMG LW spectral band are used for land and sea surfaces and are based on Huang et al. (2016).

### 4.3. 1D radiative transfer modelling

The 1D RT models in RRTMG are meant to be applied to layered atmospheres with variability of optical properties in the vertical only. As RRTMG was designed for use in large-scale models, it comes with algorithms that address

unresolved horizontal fluctuations in cloud water content and cloud overlap. These algorithms are not needed for EarthCARE because RRTMG will be applied to individual JSG columns resolved at ~1 km resolution; i.e., entirely cloud-free or -filled.

The LW transport solver in RRTMG performs flux calculations for a single diffusivity angle with an adjustment for profiles that contain large $H_2O$ vapour contents. It is an emissivity model that neglects scattering by all atmospheric

constituents. Its SW solver employs the multi-layer delta-Eddington two-stream approximation (Wiscombe 1977), which accounts for multiple scattering but, as with the LW solver, has well-documented conditional limitations for aerosol and cloud conditions (e.g., Li and Ramaswamy 1996; Barker et al. 2015a). Nevertheless, due to RRTMG's widespread use at the time of writing, it is used for EarthCARE with a minimum of alterations so as to be consistent with other current applications.

There are three applications of the 1D SW and LW RT models to each valid JSG column along the retrieved cross-section. The first, denoted as "all-sky", uses the full retrieved profiles. Second is "clear-sky" where clouds are re-moved leaving molecules and aerosols. The third application is "pristine-sky" in which clouds and aerosols are removed leaving just the molecular atmosphere.

### 4.4. 3D radiative transfer modelling

Monte Carlo solutions of the 3D RT equation are used to calculate both SW and LW fluxes and radiances. This represents a break from, and advancement over, previous satellite missions that exclusively used 1D RT solvers. The 3D RT models are discussed in the following subsections.

### 4.4.1. SW radiation

Solar fluxes and radiances are computed by a local estimation-based Monte Carlo algorithm (Marchuk et al. 1980;

Barker et al. 2003). It is discussed here in general terms, except for aspects that have not been published or were designed specifically for EarthCARE.



Unlike the 1D RT models that act on individual columns, 3D RT models require collections of columns. Photons get injected uniformly across $D^+$ that are expected to be at most ~60 km along-track by ~30 km across track (see *Figure 2*). Cosine of solar zenith angle $\mu_0$ is uniform over $D^+$ and set by its central pixel. Total numbers of inject-

ed photons per domain are to be determined, as they depend on computational resources, acceptable Monte Carlo sampling noise for either fluxes or radiances, and areal extents of individual $D^+$. Number of photons injected per spectral band is proportional to the weight associated with quadrature points in RRTMG's CKD model.

Each atmospheric cell has a spectral cumulative extinction vector whose entries for attenuating constituents are ordered, for efficiency, as: ice cloud; liquid cloud; Rayleigh scatters; absorbing gases; aerosols; and rain. When an

interaction between an attenuator and a photon takes place, a uniform random number between 0 and 1 is generated, the extinction vector is searched sequentially thus setting the attenuator, and its single-scattering properties establish whether absorption or scattering takes place (cf. Barker et al. 2003). When a scattering event occurs, a fraction $1-\omega_0$ of the photon's weight goes into local heating. What remains continues or exits out the top.

At each scattering event, the probability of photons being redirected toward a BBR telescope is determined using

$p(\mu)$. Transmittance through total optical depth between scattering event and satellite sets the probability of scattered photons getting to the satellite; as this distance is large and the telescope's aperture small, any path devia-tion is assumed to result in undetected photons. These contributions are summed to produce final estimates of BBR radiances.

The local estimation method runs into trouble when photons travelling directly toward a telescope undergo a scatter-

ing event by cloud particles whose $p(\mu)$ have sharp diffraction peaks (Iwabuchi 2006). Such rare contributions are valid, but they catastrophically elevate uncertainties, which are difficult to counter with large numbers of "typical" contributions when number of injected photons is small, as for EarthCARE. A simple way to help, without impacting fluxes and HRs, is to use the tabulated exact $p(\mu)$ to determine all photon forward trajectories but only those radiance contributions from the first $N_{\mathrm{Mie}}$ scattering events by cloud particles. Thereafter, the blunt-nosed

$p_{HG}(\mu;g)$ is used to compute radiance contributions (see Barker et al. 2003).

The rationale behind this approximation is that low-order scatterings that contribute to BBR radiances come largely from $p(\mu < 0)$, and because they do not spike radiances, several of them are allowed so as to capture details of $p(\mu)$. For optically thin clouds there will be few scattering events and so calls to $p_{HG}(\mu; g)$ might be rare. For thicker clouds, however, after ~3 scatterings photons will have had a fair chance of being redirected onto upward

travelling trajectories that can spike radiances. EarthCARE uses $N_{\mathrm{Mie}} = 4$ for, as shown in section 5.2, it strikes a balance between bias and random radiance errors (Barker et al. 2003).

When a photon arrives at the surface, it undergoes Lambertian reflection for albedo $\alpha_s$ with $1 - \alpha_s$ of its weight removed and added to *net* surface irradiance. The probability of being reflected toward a BBR sensor goes according to Lambertian for land, ice and snow, and Cox and Munk (1956) for open water (see section 4.2).

A unique, memory saving, aspect of EarthCARE's SW and LW 3D RT models is that the 3D atmosphere never appears explicitly in them. This is because all columns in $D^+$ exist along the retrieved cross-section; optical properties of columns off this plane come from a *donor* column in it, as dictated by ACM-3D's scene construction algorithm (Barker et al. 2011; Qu et al. 2022).

### 4.4.2. LW radiation

Longwave radiances are computed using the backward Monte Carlo technique (Walters and Buckius 1992; Modest 2003). This approach is very efficient at computing radiances with Cole's (2005) implementation used for EarthCARE. Much of the code resembles that of the SW Monte Carlo, and so discussion is focused on its unique aspects.

Unlike the SW Monte Carlo, photons are not injected uniformly onto the top of $D^+$ since the domain itself is the source. Rather, reciprocity of paths from an emission source to a sensor is assumed to hold (Case 1957). Hence,

photons get traced back from the top of the assessment domain to their source of emission where the contribution to radiance is computed using local temperature and optical properties. This process is repeated for each point at the in the assessment domain and radiance view angle. To reduce the number of rays traced, which is often the main computational expense, rather than trace a unique ray for each quadrature point in the CKD model it is assumed that scattering optical properties are the same for all quadrature points in a given wavelength interval.





For a given wavelength interval in the CKD model a band-representative photon path is traced backward from the top of the domain to determine a scattering path that can be related to each photon injected for each quadrature point in the band. The photon travels in straight through the domain until it has accumulated sufficient scattering optical depth to scatter in the atmosphere or scatter due to an interaction with the surface. Scatter within the atmosphere is determined based on the cumulative distribution of scattering extinction; similar to that in the SW algorithm. For

each quadrature point in CKD wavelength interval a random number is determined which sets the optical depth that must be accumulated to have an absorption event. Absorption optical depth is accumulated along the path until the photon undergoes an absorption event at which point $(1-\omega_0)B(T)$ is added to the radiance, where $B(T)$ is integrated Planck function, and $T$ is temperature. If, however, the photon is absorbed by the surface, radiance is incremented by $(1-\varepsilon)B(T_s)$, where $\varepsilon$ and $T_s$ are surface emissivity and temperature.

Upward thermal flux at a, potentially variable, reference height is also computed. This is done using a method similar to that used for radiances. The main difference being the selection (i.e., random generation) of the direction of each ray injected into the domain from the reference height. Once the ray direction is selected, accumulation of emission contributions is the same as it is for radiances.

### 4.4.3. Estimation of Monte Carlo uncertainty

For a fixed domain, 1D RT models yield single deterministic solutions. Monte Carlo algorithms, however, yield a sample from a distribution. In general, the breadth of the distribution, or Monte Carlo uncertainty, depends on the number of photons per sample, the variable being diagnosed, and the geometric and optical properties of the field.

Monte Carlo uncertainties are estimated by explicitly producing $M$ samples of a random variable $x$, each using $N_s$ photons/simulation and initialized with a unique, uniformly distributed, random number. Estimated population mean

is simply

$$\hat{\mu}_x(M,N_s) = \frac{1}{M}\sum_{m=1}^{M} x(m,N_s),$$

(12)

where $x(m,N_s)$ is the $m^{\text{th}}$ realization of $x$; i.e., estimates for the $m^{\text{th}}$ simulation. From the central limit theorem,





$$\lim_{M \to \infty} p\left[ \left| \frac{\hat{\mu}_x(M, N_s) - \mu_x}{\frac{\sigma_x(M, N_s)}{\sqrt{M}}} \right| \le a \right] = \frac{1}{\sqrt{2\pi}} \int_{-a}^{a} e^{-u^2/2} du, \tag{13}$$

where $\mu_x$ and $\sigma_x$ are mean and standard deviation of the population from which samples are drawn. Letting

$\hat{\sigma}_x(M, N_s)$ be an estimate of $\sigma_x$ based on $M$ samples, Monte Carlo "uncertainty" is defined as one standard

deviation under a Gaussian distribution of samples. This amounts to setting $a = 1$ in (13), and implies that after $M$

realizations, $\hat{\mu}_x$ has a 68% chance of lying in

$$\left[ \hat{\mu}_x(M, N_s) - \frac{\hat{\sigma}_x(M, N_s)}{\sqrt{M}}, \hat{\mu}_x(M, N_s) + \frac{\hat{\sigma}_x(M, N_s)}{\sqrt{M}} \right], \tag{14}$$

making for an uncertainty of

$$\hat{\rho}_x(M, N_s) \approx \pm \frac{\hat{\sigma}_x(M, N_s)}{\sqrt{M}}. \tag{15}$$

This is an approximation as it arises only as $M \to \infty$. As $M$ increases, estimates of $\hat{\sigma}_x$ stabilize; they do not go to

zero.

## 5. Results

This section's main purpose is to showcase a small sample of EarthCARE's radiation products; some of which get

utilized directly for radiative closure assessment as will be reported in a later study. Results are shown using only

ACM-CAP data; corresponding results for ACM-COM's composites are qualitatively the same.

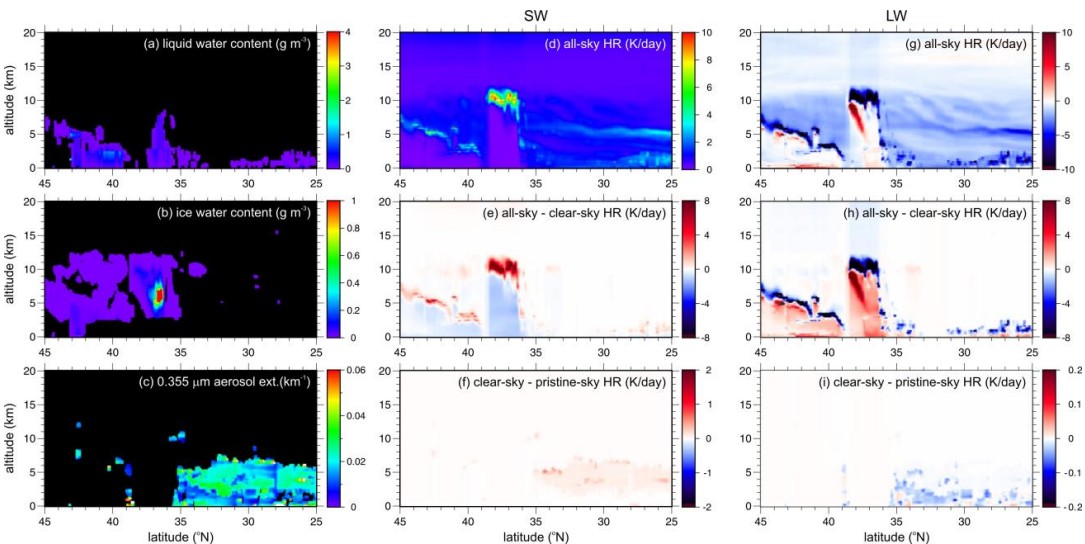

**Figure 4:** (a) Profiles of domain-average cloud liquid water content, (b) ice water content, and (c) aerosol extinction coefficient for 21 km-long assessment domains, for the *Halifax* frame, as inferred by ACM-CAP's synergistic algorithm. (d) Corresponding domain-average, all-sky SW broadband heating rates computed by RRTMG's 1D RT model. (e) Difference between HRs shown in (d) and those computed by RRTMG for clear-sky conditions. (f) As in (e) except these HR differences are for clear-skies and pristine-skies. (g), (h), and (i) are as in (d), (e), and (f), respectively, except these are for LW broadband heating rates.

## 5.1. RRTMG 1D fluxes: Pristine-, clear-, and all-sky

As discussed in section 2, broadband flux and heating rate profiles for all admissible L2 columns are computed by RRTMG's SW and LW 1D RT models. The left column of *Figure 4* shows ~2,200 km of cloud and aerosol properties retrieved by ACM-CAP's synergistic algorithm (Mason et al. 2022). These results pertain to 21 km-long non-overlapping assessment domains near the central of the *Halifax* test frame. The middle column shows corresponding SW all-sky HRs and differences between all-sky HRs and clear-sky HRs (cloud radiative effect: CRE), and clear-sky HRs and pristine-sky HRs (aerosol direct effect: ADE). Aside from the usual 1D RT features, such as large SW heating near cloudtop and much smaller values below relative to clear-sky, the only peculiarity is the fairly strong heating at ~5 km altitude in the south-end. This is due to an elevated layer of water vapour. The vast majority of minor heating due to aerosol is from continental pollution that overrides sea salt.

The rightmost column in *Figure 4* is like the middle column but it shows results for LW HRs. As expected, there is strong cooling in the upper 1 - 2 km, or so, of clouds, little net heating or cooling below, and general cooling from





cloudless-skies. LW CREs are generally stronger than in the SW and exhibit strong cooling near all cloudtops and warming in clouds, when they are of sufficient vertical extent. LW ADEs are an order of magnitude smaller than

their SW counterparts, and manifest themselves as cooling just beneath their SW warming counterparts.

To demonstrate what will be available in the ACM-RT archive, *Figure 5* shows TOA CRE, ADE, and some integrated cloud and aerosol properties that correspond with *Figure 4*. Some noteworthy points here are SW CRE reaching ~300 W m$^{-2}$ at $\mu_0 \approx 0.3$ due to clouds near $41°$N with large cloud water paths (CWP), LW CRE reaching -100 W m$^{-2}$ near $37°$N due to supercooled liquid aloft, and weak ADE (~ 10 W/m2 in the SW and less than -1 in the LW)

stemming from aerosol optical depth, at 0.355 μm, being at most 0.2. Aside from this, there is very little to comment on in these plots; they serve to demonstrate what will be available in the ACM-RT archive.

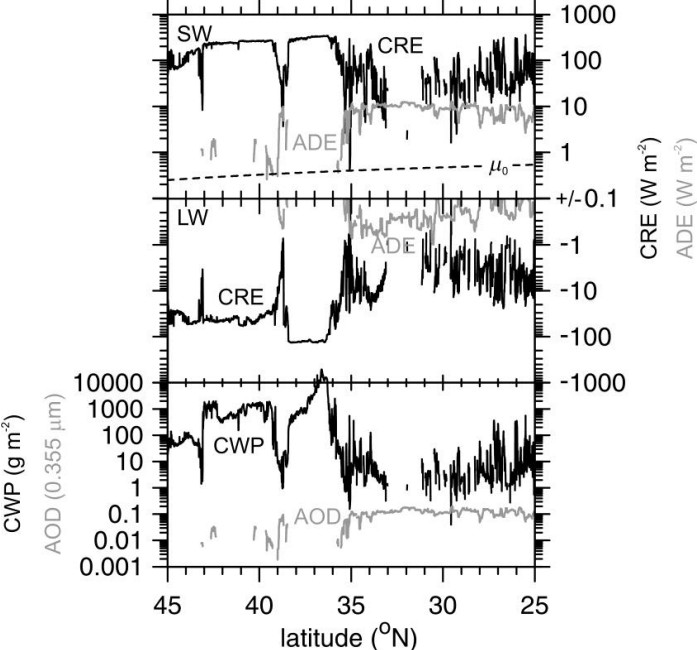

**Figure 5:** Top panel: Cloud radiative effect (CRE) and aerosol direct effect (ADE) as functions of latitude for broad-
band SW at an altitude of 20 km for 21 km-long assessment domains as shown in *Figure 4*. $\mu_0$ is cosine of solar zenith angle. Middle panel: As in top panel except it is for broadband LW. Lower panel: Assessment domain-average cloud water path (CWP) and aerosol optical depth (AOD).





### 5.2. On the benefits of employing 3D RT models

As mentioned above, one of EarthCARE's notable advancements over prior like-missions is operational use of both

1D *and* 3D RT models. The decision to use 3D RT models was fuelled by myriad studies that show systematic

differences between 1D and 3D treatments of RT, especially for cloudy atmospheres at solar wavelengths. Results

shown in this subsection help justify the computational expensive of using 3D RT models operationally.

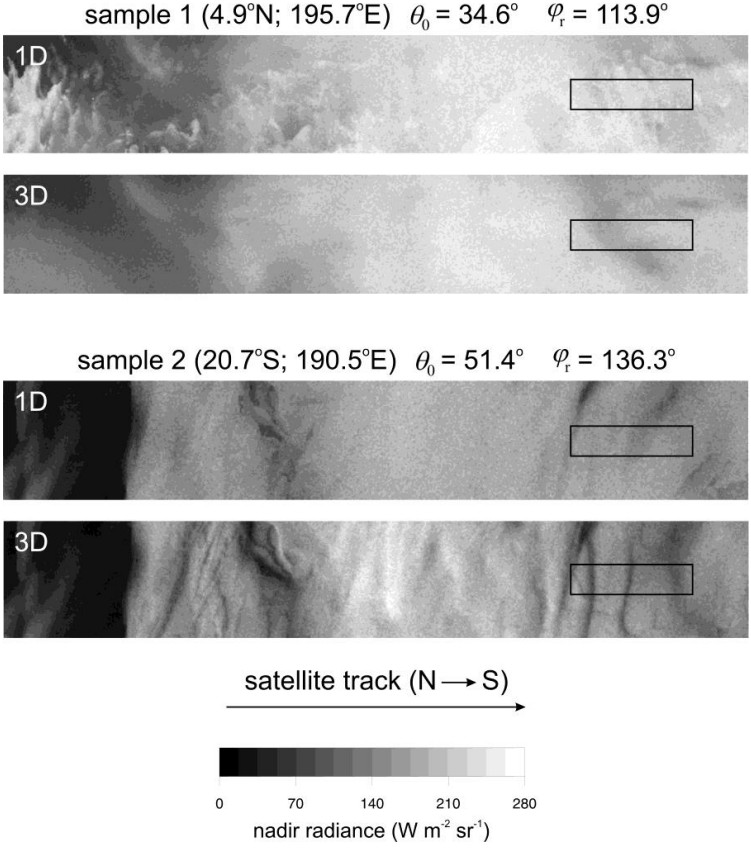

**Figure 6:** Nadir broadband SW radiances for two sample regions in the Hawaii frame; both regions measure 128 km
along-track by 20.25 km across-track. Small rectangles indicate a 5 x 21 km assessment domain. Central values of
latitude and longitude are listed along with $\theta_0$ and $\varphi_r$ (measured clockwise from the satellite's tracking direction).
Labels 3D and 1D indicate RT model dimensionality using horizontal grid-spacings of 0.25 km and $10^6$ km.

Before getting to results that apply strictly to EarthCARE, consider a detailed view of the impact of neglecting multi-

dimensional RT. *Figure 6* shows nadir SW radiances simulated by a Monte Carlo RT model (Villefranque et al.


2019) for two stretches of the *Hawaii* test frame, each measuring 128 km along-track by 20.25 km across-track (Qu et al. 2023). The 3D RT simulation used horizontal grid-spacing $\Delta x = 0.25$ km while its 1D rendition used $\Delta x$ set arbitrarily large. Hence, differences in their radiances stem entirely from the dimensionality of the RT solution. For 445 this demonstration, the number of photons per column was 4,096, which is, on an areal density-basis, several times larger than what will be used operational for the EarthCARE mission.

These images display the varied and complicated ramifications on radiances when 1D RT modelling theory is assumed to apply. For sample 1, 1D radiances show much variability and sharp contrasts relative to their 3D counterparts; off-nadir views (not shown) look much the same. This region is blanketed by thick overcast ice cloud, which at 450 $\Delta x = 0.25$ km, act to diffuse upwelling radiation, thus blurring localized reflection from low-level intermittent liquid clouds (e.g., Diner and Martonchik 1984). When 1D RT is affected by setting $\Delta x$ large, however, flow of radiation is confined to the vertical and the sharp features of liquid clouds remain intact regardless of altitude.

On the other hand, sample 2 has mostly low-to-mid-level liquid clouds and shows, due in part to large $\theta_0$, the more familiar differences between 3D and 1D RT (e.g., Barker et al. 2017). In particular, 1D radiances lack texture, whilst 455 their 3D counterparts exhibit much contrast due to shadowing and cloud-side illumination. Note, however, that imagery for thin liquid clouds at the northern edge of the sample depend little on $\Delta x$. This is because reflected photons undergo small numbers of scattering events and thus tend to exit clouds close to where they enter.





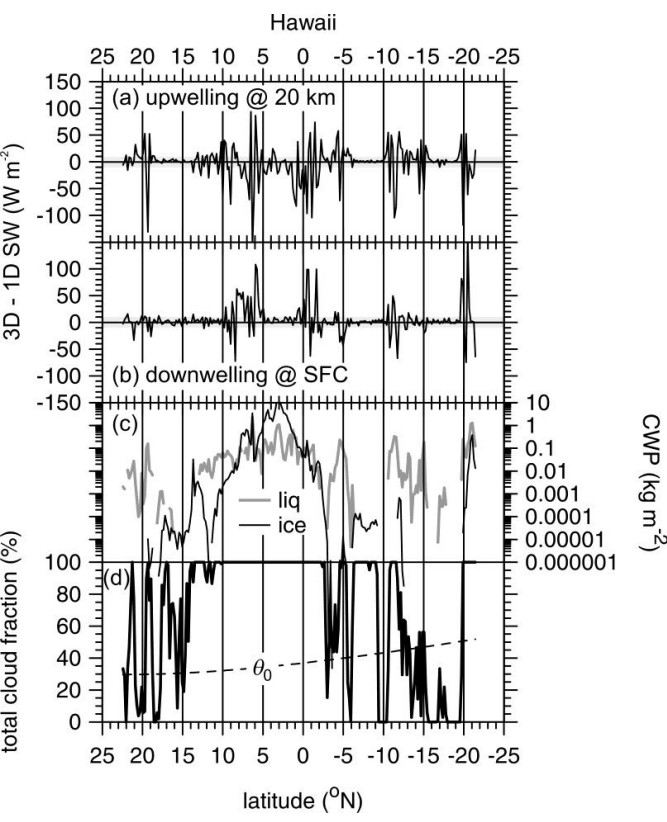

**Figure 7:** (a) Difference between upwelling SW fluxes at an altitude of 20 km as predicted by 3D and 1D RT models for 5 x 21 km assessment domains of the Hawaii frame. Shaded area indicates EarthCARE's goal of $\pm 10$ W m$^{-2}$. (b) As in (a) except this is for SW surface irradiance. (c) Mean liquid and ice cloud water paths for the Hawaii frame's 5 x 21 km domains. (d) Corresponding total cloud fraction and solar zenith angle for the same assessment domains.

Consider now differences one can encounter in applications to EarthCARE retrievals. *Figure 7* shows differences between 3D and 1D RT modelled SW broadband upwelling fluxes at 20 km and surface irradiances for 5 x 21 km assessment domains across the *Hawaii* frame using ACM-CAP cloud properties. Values for 3D and 1D RT are from the Monte Carlo model using $\Delta x = 1$ km and arbitrarily large $\Delta x$, respectively. Each simulation used $2.5 \times 10^6$ photons, which is likely much larger than what will be used operationally throughout the mission. For almost cloud-free skies, thin ice cloud-only with ice water path IWP < 0.01 kg m$^{-2}$, and very thick clouds with CWP > 0.5 kg m$^{-2}$, differences are well within $\pm 10$ W m$^{-2}$ for fluxes at both levels. Clearly, under these conditions SW photon trajectories are characterized by either extremely small or large numbers of scattering events with cloud particles for both 1D



and 3D RT. For the majority of other cloud conditions, however, especially with CWP in the vicinity of ~0.1 kg m$^{-2}$, differences can be much larger than $\pm 30$ W m$^{-2}$, which far exceeds EarthCARE's goal (ESA 2001; Illingworth et al.

2015; Eisinger et al. 2022). The implication being that many attempts to perform a radiative closure assessment on EarthCARE's retrievals will be doomed to failure if 1D RT models are adhered to.

*Figure 8* shows cumulative frequency distributions of the differences shown in *Figure 7* for several ranges of total cloud fraction $A_c$. For upwelling fluxes at 20 km with $A_c < 0.25$, median differences are all close to zero. The same goes for 3D - 1D mean-bias errors (MBEs) as listed in *Table 1*. Differences tend to be distributed more or less

symmetrically about zero with occasional large differences, exceeding $\pm 50$ W m$^{-2}$, enhancing root mean-square errors (RMSEs) as $A_c$ increases (see *Table 1*) relative to the 16- and 84-percentiles of the distributions, which can be gleaned from the graphs.

There are at least two interesting points to these plots that involve extremal cloud conditions. First, 3D - 1D can be expected to be maximized for overcast domains, which implies that the geometry of overcast clouds is often anything

but approximately plane-parallel and homogeneous (cf. Hogan et al. 2019). Second, for assessment domains $D$ with $A_c = 0$, 3D - 1D values for upwelling flux at 20 km show a tendency to be positive on account of contributions from clouds in the surrounding buffer-zone (see *Figure 2*).

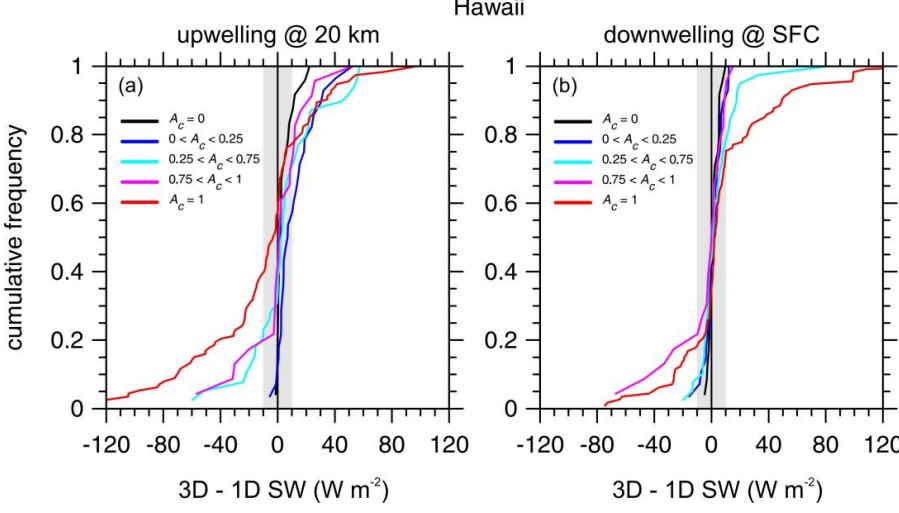

**Figure 8:** (a) Cumulative frequency distributions for differences between 3D and 1D Monte Carlo RT model esti-
mates of upwelling SW flux at an altitude of 20 km for 5 x 21 km assessment domains for the Hawaii frame parti-





tioned according to assessment domain total cloud fraction $A_c$ (see *Table 1* and *Figure 7*). Shaded area indicates EarthCARE's goal of $\pm 10$ W m$^{-2}$. (b) As in (a) except these are for surface (SFC) irradiances.

**Table 1:** Mean 3D SW RT values, mean bias errors (MBEs), and root mean-square errors (RMSEs) for correspond-
ing 3D - 1D RT results (see *Figure 7* and *Figure 8*) for 5 x 21 km assessment domains for the Hawaii frame and several ranges of total cloud fraction $A_c$.

| total cld frac | | cases | upwelling flux at 20 km (W m$^{-2}$) | | | | SFC irradiance (W m$^{-2}$) | | |
|---|---|---|---|---|---|---|---|---|---|
| | | | 3D RT | MBE | RMSE | | 3D RT | MBE | RMSE |
| $A_c = 0$ | | 24 | 81.0 | 4.2 | 6.2 | | 698.0 | 1.5 | 3.6 |
| $0 < A_c < 0.25$ | | 28 | 93.5 | 12.2 | 13.8 | | 780.0 | 1.5 | 6.3 |
| $0.25 < A_c < 0.75$ | | 39 | 112.0 | 5.0 | 25.2 | | 755.9 | 4.6 | 16.1 |
| $0.75 < A_c < 1$ | | 23 | 128.0 | 1.3 | 21.6 | | 777.0 | -5.6 | 19.7 |
| $A_c = 1$ | | 113 | 395.5 | -11.5 | 41.8 | | 462.7 | 8.6 | 35.2 |

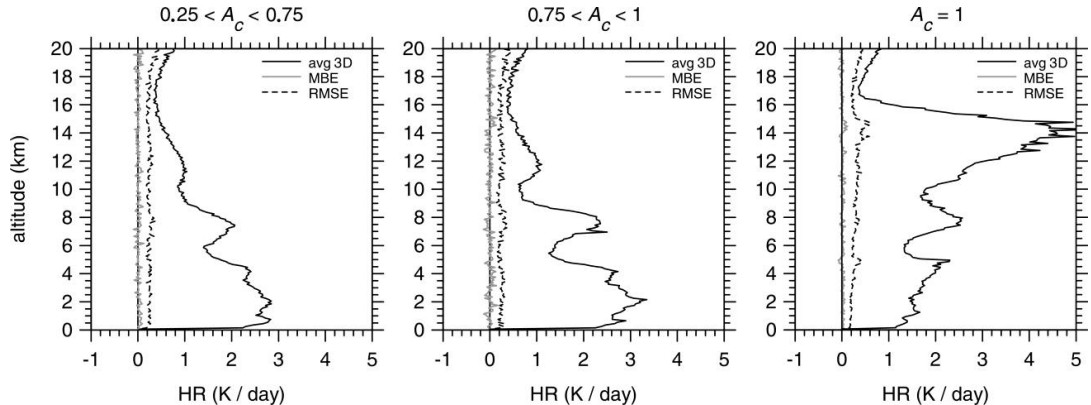

**Figure 9:** Mean 3D RT SW heating rate (HR) profiles for 5 x 21 km assessment domains for the Hawaii frame
partitioned according to assessment domain total cloud fraction $A_c$ (see *Figure 7*). Also shown are mean bias errors (MBEs) and root mean-square errors (RMSEs) between 3D and 1D RT models. Numbers of cases per $A_c$ range are listed in *Table 1*.

*Figure 9* shows that SW HR differences between 3D and 1D RT for the *Hawaii* frame's 5 x 21 km assessment
domains are much less dramatic than those seen in *Figure 7* and *Figure 8* for boundary fluxes. At all altitudes and





ranges of $A_c$, MBEs are essentially zero and close in magnitude to Monte Carlo uncertainties for $2.5 \times 10^6$ photons. There are several reasons why RMSE values are ~10x larger than Monte Carlo uncertainties, and only increase slightly as $A_c$ increases. There are the obvious differences due to cloud side illumination, shadowing, and photon entrapment (Hogan et al. 2019), as well as impacts on flux profiles for 3D RT due to out-of-domain sources and

sinks of photons; i.e., clouds outside $D$, but still in $D^+$, that cast shadows or scatter radiation into $D$.

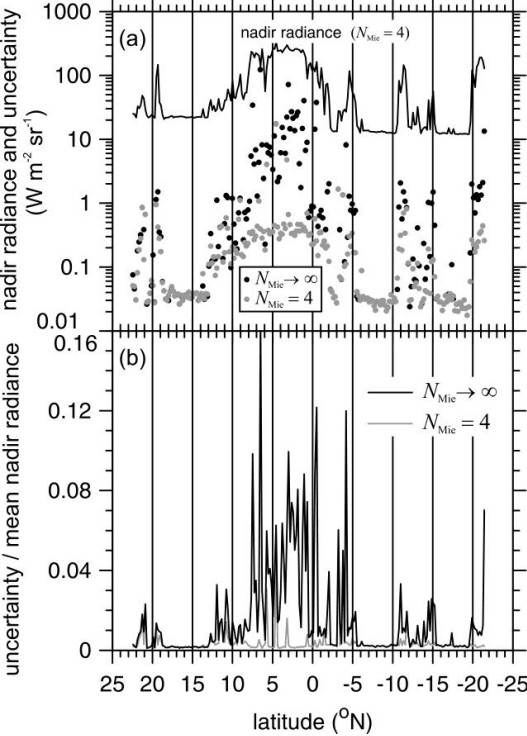

Figure 10: (a) Line is 3D RT nadir broadband radiances using $\Delta x = 1$ km when reverting to the Henyey-Greenstein phase function $p_{HG}$ after $N_{\mathrm{Mie}} = 4$ cloud particle scattering events for 5 x 21 km assessment domains of the Hawaii frame. Dots are Monte Carlo uncertainties when $p_{HG}$ is never used ( $N_{\mathrm{Mie}} \to \infty$ ) and when it is used after

4 cloud scattering events ( $N_{\mathrm{Mie}} = 4$ ). (b) Using data in (a), Monte Carlo domain-average uncertainties relative to mean values for both values of $N_{\mathrm{Mie}}$. Each domain received $2.5 \times 10^6$ photons.

There is the possibility that radiative closure assessments of cloud and aerosol retrievals could (i.e., should) use broadband radiances rather than fluxes. There are reasons both for and against this. For instance, off-nadir BBR

radiances offer powerful assessments due to their week correlation, relative to nadir BBR radiances, with MSI

radiances that are used for some retrievals. They can, however, arise from attenuators outside the domain being

assessed (see Barker et al 2015b). On the other hand, all of EarthCARE's performance goals are in terms of BBR

fluxes, which will be estimated regularly by tailor-made algorithms (Velázquez-Blázquez et al., 2022a) despite

adding, at times substantial, uncertainty at the last step of EarthCARE's processing chain.

Regardless, SW BBR radiances will be estimated throughout the mission. *Figure 10* shows nadir values for the

*Hawaii* frame's assessment domains using $2.5 \times 10^6$ photons per assessment domain and $\Delta x = 1$ km. It also shows

relative Monte Carlo uncertainties for $N_{\mathrm{Mie}} = 4$ and $N_{\mathrm{Mie}} \to \infty$. As $2.5 \times 10^6$ photons / domain is likely to be

more than routine operations can afford, uncertainties for $N_{\mathrm{Mie}} \to \infty$ could be substantially larger than those shown

here. This would render them useless for most assessments. While use of $N_{\mathrm{Mie}} = 4$ will help, as is evident for the

thick clouds between $0°$ to $10°\mathrm{N}$ and near $20°\mathrm{S}$, it will foster errors in radiances themselves. Two options are

being considered: i) use radiances, instead of fluxes, for assessments when their relative Monte Carlo uncertainties

are less than some specified value (e.g., 0.01; see *Figure 10*); and ii) unbiased variance reduction methods (e.g.,

Iwabuchi 2006).

As is well known, flux and radiance differences between 3D and 1D treatments of RT for LW radiation are usually

much smaller than those for SW radiation (e.g., Ellingson and Takara 2005; Cole et al. 2005; Hogan et al. 2016;

Fauchez et al. 2017). *Figure 11* shows the LW counterpart of the upper panel in *Figure 7*. When differences go

beyond $\pm 10$ W m$^{-2}$, they do so along with corresponding large differences in SW fluxes; typically for overcast skies

with CWP ~0.1 kg m$^{-2}$. As shown in *Figure 12*, ~5% of overcast cases exhibit 3D fluxes that are less than their 1D

counterparts by more than 10 W m$^{-2}$. For these domains, CWPs are small relative to their neighbouring domains.

This demonstrates a difficulty when interpreting "fluxes" for $5 \times 21$ km domains: at 20 km altitude, fluxes for 3D

RT can be influenced substantially by adjacent cloudier domains. *Table 2*, however, shows that 3D and 1D fluxes

usually differ by less than $\pm 1$ W m$^{-2}$ which is on the order of the Monte Carlo uncertainty for these calculations,

roughly 0.2 W m$^{-2}$.



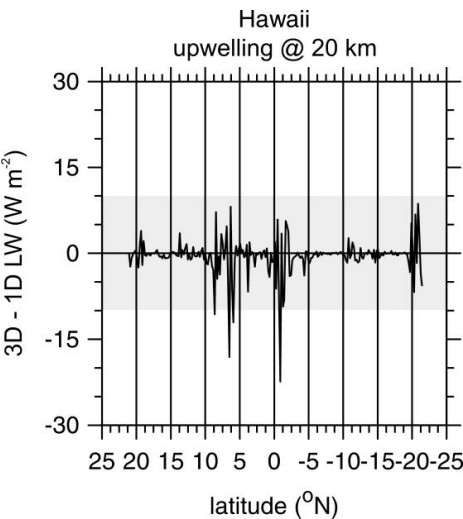

Figure 11: Difference between upwelling LW fluxes at an altitude of 20 km as predicted by 3D and 1D RT models for 5 x 21 km assessment domains of the Hawaii frame. A positive value means that 3D upwelling flux exceeds its 1D counterpart. Shaded area indicates EarthCARE's goal of $\pm 10$ W m$^{-2}$.

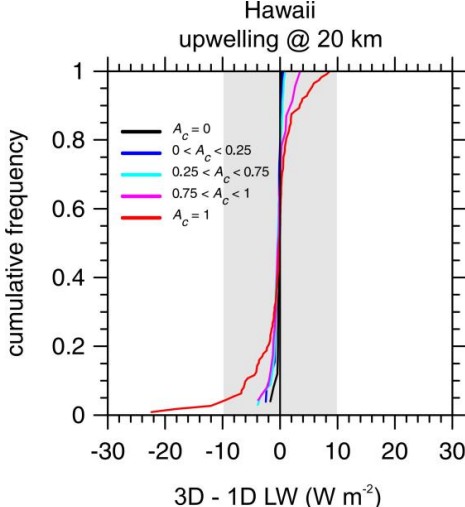

Figure 12: Cumulative frequency distributions for differences between 3D and 1D RT model estimates of upwelling LW flux at an altitude of 20 km for 5 x 21 km assessment domains for the Hawaii frame partitioned according to assessment domain total cloud fraction $A_c$ (see *Table 2*). Shaded area indicates EarthCARE's goal of $\pm 10$ W m$^{-2}$.



**Table 2:** Mean 3D LW RT values, mean bias errors (MBEs), and root mean-square errors (RMSEs) for corresponding 3D - 1D RT results (see *Figure 7* and *Figure 8*) for 5 x 21 km assessment domains for the Hawaii frame and several ranges of total cloud fraction $A_c$.

| total cld frac | | cases | | upwelling flux at 20 km (W m$^{-2}$) | | |
|---|---|---|---|---|---|---|
| | | | | 3D RT | MBE | RMSE |
| $A_c = 0$ | | 25 | | 285.4 | -0.2 | 0.4 |
| $0 < A_c < 0.25$ | | 26 | | 289.2 | -0.5 | 0.7 |
| $0.25 < A_c < 0.75$ | | 34 | | 286.0 | -0.5 | 1.0 |
| $0.75 < A_c < 1$ | | 23 | | 287.5 | -0.2 | 1.5 |
| $A_c = 1$ | | 112 | | 208.9 | -0.7 | 4.3 |

## 6. Summary

The EarthCARE satellite mission's objective is to retrieve profiles of aerosol and water cloud physical properties from measurements made by its cloud-profiling radar (CPR), backscattering lidar (ATLID), and passive multi-spectral spectral imager (MSI). While several L2a processes infer geophysical properties using measurements from a single sensor (see several articles in this special issue), EarthCARE's primary product comes from the L2b synergistic retrieval algorithm in ACM-CAP (Mason et al., 2022). These retrievals, together with other geophysical properties obtained either from pre-existing satellite data or real-time weather prediction models, are input into broadband (BB) radiative transfer (RT) models that predict radiances, and fluxes, commensurate with measurements made, and inferred from, independently by EarthCARE's BB radiometer (BBR). The scientific goal is that modelled and "observed" BB fluxes differ, on average, by less than $\pm 10$ W m$^{-2}$.

This report described the RT models used for EarthCARE and their products, which together comprise the ACM-RT process. Shortwave (SW) and longwave (LW) flux and heating rate (HR) profiles are computed by the 1D solver-based RRTMG for each ~1 km nadir column of inferred properties. In addition to the 1D RT models, which are ubiquitous to almost all operational and research satellite missions, EarthCARE is the first to employ 3D RT models operationally. Both SW and LW models will compute radiances for the BBR's three viewing directions, with the SW





model also computing BB flux and HR profiles. The 3D LW model produces only upwelling fluxes at a variable reference level as dictated by the BMA-FLX process (Velázquez-Blázquez et al., 2022a). All 3D RT products are averages over $5 \times 21$ km "assessment domains" that are constructed in the ACM-3D process (Barker et al. 2022) using a radiance mapping algorithm (Barker et al. 2011) and MSI data.

When the ACM-CAP process runs successfully, its retrievals are operated on by the RT models. Failing this, the RT models are applied to "composite" atmospheric profiles generated in the ACM-COM process by combining L2a retrievals from individual sensors. Usually, this involves filling grid-cells with retrievals from either CPR *or* ATLID data. When two L2a estimates exist for a cell, the one with the least relative uncertainty is selected. ACM-COM also prepares either ACM-CAP or composites for use in RT models by bringing together information about atmospheric state and surface optical properties. Regardless of what atmosphere is used, nadir profiles are broadened across-track by mapping indices from ACM-3D in order to create 3D domains for the 3D RT models to use. A subset of ACM-RT's products are passed forward to the ACMB-DF process where a "radiative closure assessment" executes in an attempt to quantify the likelihood that EarthCARE's goal has been achieved.

Data from the EarthCARE test frames (Qu et al. 2022; Donovan et al. 2022) were used to demonstrate some of the products to be expected from ACM-COM and ACM-RT. In several respects, products associated with the 1D RT models resemble closely those available via the CloudSat mission (e.g., L'Ecuyer et al. 2008). The most notable extension is that ACM-RT will be reporting continuous cloud and aerosol radiative effects based on 3D RT model results.

The majority of the results reported here (see section 5.2), however, had to do with the benefits expected from operational application of 3D RT models. The ACM-RT process is the most computationally intensive one in Earth-CARE's processing chain. While a significant amount of computer time is required by both of the 1D RT models and 3D LW RT model, the lion's share of ACM-RT's allocated time is consumed (inevitably entirely) by the 3D SW RT model. Its voracity is such that only a portion of a frame's available assessment domains will be operated on; the expectation being, however, that sufficient numbers of samples will be realized over the duration of the mission. This is primarily because of the large number of photons that have to be injected into the Monte Carlo RT model in order to produce flux and radiance estimates with uncertainties small enough to realize beneficial radiative closure assessments in the ACMB-DF process (Barker et al. 2022). The most demanding product is off-nadir radiances. Despite

limited attempts to reduce Monte Carlo variance, it still might be necessary to limit computation to nadir radiance and one off-nadir radiance. Moreover, there is still the option to forego the radiances, thereby increasing greatly the number of assessment domains with 3D SW RT fluxes, and use just fluxes in ACMB-DF. This will be determined during EarthCARE's commissioning phase.

If results presented in *Table 1* and *Figures 5* through *7* can be taken as representative, operational use of SW 3D RT
modelling will be well-worth the heavy computational load to be incurred. This is because differences between 3D and 1D RT values of upwelling fluxes and radiances can be either positive or negative (cf. Hogan et al. 2019) and can often exceed EarthCARE's goal of being able to, effectively, retrieve properties to within $\pm 10$ W m$^{-2}$; of course the warning here is that continued reliance on 1D RT models would amount to a heightened frequency of radiative closure assessments being unwittingly nullified.

**Data availability**

The EarthCARE Level-2 demonstration products the ACM-COM products discussed in this paper are available from https://doi.org/10.5281/zenodo.7117115 (van Zadelhoff et al., 2022) as are "operational" ACM-RT output. Specialized ACM-RT calculations presented in this paper, e.g., with increased photon count, and radiative transfer calculations are available from https://doi.org/10.5281/zenodo.7272662 (Cole et al., 2022).

**Competing interests**

The authors have no competing interests to declare.

**Author contributions**

HWB drafted the manuscript and developed the ACM-RT and ACM-COM algorithms and the 3D solar radiative transfer model used in ACM-RT. JNSC developed the ACM-RT software, developed the 3D thermal radiative
transfer model and performed all ACM-RT calculations used in the manuscript. ZQ developed ACM-COM algorithms and software plus aerosol look-up table for ACM-RT. NV contributed to the development and testing of



ACM-RT and performed independent 3D solar radiative transfer calculations. MS integrated the 1D radiative transfer model into ACM-RT. All authors were involved in development of ACM-RT and ACM-COM and contributed material and/or test to the manuscript.

**Acknowledgements**

We thank Tobias Wehr and Michael Eisinger for their continuous support over many years and the EarthCARE developer teams for valuable discussions in various meetings.

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
