# Peer review of "Broadband Radiative Quantities for the EarthCARE Mission: The ACM-COM and ACM-RT Products"

_Atmospheric Measurement Techniques, 2022_

## Author Comment (AC1)

Response to Referee #1

We thank the referee for their review.  Below are the original comments in *italics* with our responses in normal text.

*This well-written and well-presented paper is part of a collection describing algorithms and products of the EarthCare mission, and probably makes more sense when read in conjunction with (some) of these other papers. I'm not sure how a paper of this type should be reviewed: is it more about the clarity and quality of the presentation or are criticisms of the algorithm appropriate at this stage meaningful? I guess it's more about the former rather than the latter, and fortunately I don't have much to say about the latter either even if I were to focus on that aspect. So this review is more about questions on matters that were somewhat unclear to me, and may also puzzle other readers not so familiar with the mission or the accompanying papers.*

We agree that this contents of this manuscript makes more sense when combined with the rest of the manuscripts describing the EarthCARE.  It is a large and complex mission.  We thank the referee for suggestions to clarify the presentation.

 *Questions such as:*

*-- What is a "frame"? A citation is provided, but can the concept be explained in couple of sentence. Where does the "truth" in the frame come from since there aren't yet EarthCare retrievals of either the L2a or L2b variety? I'd assume some sort of cloud resolving model, (GEM? Fig. 3). So, there are model-generated cloud fields, forward calculations with instrument simulators, and then a cloud retrieval by applying an inversion algorithm on the simulated signals? Does this process corrupt at all the closure effort? What if the RT calculations were applied directly to GEM fields rather than retrevals from the GEM fields, would such an experiment be useful?*

The text has been adjusted to indicate what is a "frame".  Basically it is a 6200 km section of the EarthCARE orbit.

The "truth" is as you describe.  Forward radiative transfer calculations are applied to high resolution numerical weather prediction (GEM model) output, which are then used as input to instrument simulators.  The simulator results are used as input to the entire EarthCARE algorithm process, which are then used by the ACM-RT processor.  Radiative closure is performed by comparing output from ACM-RT and processed BBR "observations" consistent with what will be done when observations are available from EarthCARE.  Having the high resolution model fields from GEM does afford us the ability to apply the RT calculations directly to the original data.  This is indeed a useful experiment as it allows performing a "perfect" radiative closure in which differences should be attributable to instruments and retrievals.  Such an analysis is not included in this manuscript but is one of our planned activities.

*-- Why are the domains 5x21 km, what's so special about this choice?*

The size of the domain for the radiative closure is ~100 km$^2$ which was driven in part by performance of the BBR which had its nominal performance requirements defined for this sized domain. Instead of using a 10 km by 10 km domain, we decided to use a domain that is larger orbit and smaller across orbit. This is a balance between a domain that is far from the track of the active sensors, for which the scene construction algorithm would have increasing impact on the results, and a narrow but long domain that would make interpretation of the closure difficult. The use of 21 km along orbit rather than 20 km is due to the 7 km period of the horizontal grid used for for retrievals (Eisenger et al, 2023).

We have added text and references to the manuscript to explain this choice.

Eisinger, M., T. Wehr, T., Kubota, D., Bernaerts, and K. Wallace, 2023: The EarthCARE production model and auxiliary products. *Atmospheric Measurement Techniques*, to be submitted.

*-- Why is Dx= 0.25 km used for the experiment of Fig. 6, while Dx=1 km used later (line 468). Shouldn't the resolution of the ACM-COM or ACM-CAP retrieval only be used (BTW, do these abbreviations need to be listed somewhere, is it important to know what they stand for?)*

The results shown in Figure 6 are RT calculations applied directly to the GEM fields which has a 250 m horizontal resolution. For the EarthCARE retrievals, the data is on a 1 km horizontal grid. Since Figure 6 is used for illustrative purposes of the 3D effect and not quantitative analysis we believe that it is OK to show it at the highest possible horizontal resolution.

*-- Why are the signs of SW and LW CREs in Fig. 5 different than what we're accustomed to (negative and positive, respectively).*

We thank the referee for catching this error creating the plot in the figure. It indeed is reversed from the CREs computed using the standard definition. Figure 5 and associated text has been corrected and updated.

*-- If a radiance closure approach were to be used, what would be the criterion for "pass"?*

We do perform radiative closure using radiances but it is not discussed in this manuscript which focuses on the radiative transfer calculations used for the radiative closure. Details of the closure using radiances, including the criteria for a "pass" are discussed in a separate paper focused on the EarthCARE radiative closure processor.

*-- If closure is not satisfactory, is there some post-processing provision to "fix" the retrievals to achieve closure (I imagine such a possibly iterative revision would be complicated).*

There is not an iterative adjustment of the retrievals to improve the radiative closure in the operational processing of EarthCARE data.  However, the radiative closure results are certainly used to inform ongoing improvements of the retrievals for future reprocessing.

*-- What an obscure reference for water refractive index (Segelstein).  Are the authors aware of Platnick et al. (2020) https://doi.org/10.3390/rs12244165 where the importance of refractive indices is discussed (for inversion, not forward BB calculation).*

We are aware of the paper by Platnick et al, 2020 but the reference cited in the manuscript is consistent with the refractive indices used in the Mie calculations described in this manuscript.

*-- I don't see a shaded area in Fig. 7 even if the caption of the figure mentions one.*

It is there but admittedly, it is faint and given the vertical range of the y-axis quite close to the zero line.  We have darkened the shading so it should be more easily seen.

*-- Shouldn't the authors comment about the lack of closure possibly being i many cases due to factors other than cloud retrievals, inadequate 1D BB RT, or imperfect 3D BB RT? Like wrong assumptions and input? What if the ice models are not realistic, for example?*

We agree that the lack of closure could be affected by a range of factors both within our control, e.g., the radiative transfer modelling, and outside of our control, e.g., surface properties.  The goal of this manuscript is to highlight the importance of 3D radiative transfer using the 10 W m$^{-2}$ as a metric.  The effect of the input uncertainty on the radiative closure is discussed in the manuscript describing the EarthCARE radiative closure processor.

---

## Author Comment (AC2)

Response to Referee #2

We thank the referee for their review.  Below are the original comments in *italics* with our responses in normal text.

*The manuscript presents the broadband radiative transfer (RT) algorism for the EarthCARE mission. The 1D and 3D RT schemes are implemented. Because the 3D RT scheme will be used operationally in the satellite mission for the first time, the authors presented in this manuscript how 3D RT is important in to achieve the mission's goal for the accuracy in radiative flux. The manuscript is generally well written. I recommend this paper is published after some revisions. Below are my specific comments and questions.*

**General comments**

1. *In the manuscript, the authors used several technical terms specific to the EarthCARE project, such as "joint standard grid (JSG) column". Although they are shown with references, many of which are in the EarthCARE special issue, and they are unknown for the readers. It is helpful for the readers to be with some (even short) explanations (e.g., what to do for what purpose).*

   We have added text as and where appropriate in the manuscript to give a better sense of the EarthCARE specific technical terms.

2. *This is a similar issue as above. I miss the algorithm description of the "observed" radiative fluxes from BBR. Because the BBR actually measures the broadband radiances, the "observed" radiative fluxes would be some estimates. There is a citation to a manuscript that has not yet been published, but I could not find any further information about the paper. A brief explanation is required in the manuscript. Is the observed radiative flux accurate enough to compare with simulated radiative fluxes? As for radiative closure analysis, I support the usefulness of BBR multiangle radiance (instead of radiative fluxes).*

   The referee is correct that the "observed" radiative fluxes are estimates derived from the radiances.  The citations for the paper describing the BMA-FLX processor, detail the method using angular distribution models.  We do not believe that an explanation of the approach is necessarily required in this manuscript since the fluxes computed by the BMA-FLX processor are not used in this manuscript.  The fluxes only described to indicate the reason for the choices made for the forward radiative transfer calculations, including the calculation of fluxes at particular reference heights and the size of the radiative closure assessment domains.

   However, the fluxes from the BMA-FLX processor are used in the radiative closure, so details of the radiance to flux conversion are described in the manuscript focused on the EarthCARE radiative closure processor.  This includes questions related to the accuracy

and uncertainty of the radiative fluxes both derived from the BBR and computed in ACM-RT.

3.  *If local positive and negative 3D-1D flux differences are well cancelled in larger scales, 1D flux is enough to explain larger scale average. Averaging-scale dependence is interesting to see. Is there any plan to study that aspect in the future radiative closure studies?*

    Yes, there is a plan to look at this in future studies.  It is an interesting way to analyse the 3D effects for both domains smaller and larger than that shown in this manuscript and relevant for analysis and interpretation.  For example, to characterize the magnitude of the 3D effect in numerical weather prediction models, at the same scale or smaller, in contrast with climate models, which typically have larger scales.

4.  *I have a question regarding the 3D radiative effect: Why the goal of the EarthCARE mission is set at local (of ~100 km$^2$ domain) radiative flux accuracy? I completely agree that 3D RT is required to simulate BBR radiances and to obtain the radiative closure, while the local radiative flux at TOA (~20 km height) is not directly measured. What is the benefit to obtain local 3D flux? Is CRE evaluated for 3D RT as well? If so, it would be interesting to see how CRE is different from 1D RT counterpart.*

    The local radiative flux accuracy was defined early in the EarthCARE mission.  According to this document ESA 2001 cited in the manuscript, the size of the horizontal domain was the scale expected to the roughly the size of numerical weather prediction systems at the time of launch.  The 10 W m$^{-2}$ level of accuracy was noted to be the level needed to detect radiatively important variations in the retrieved profiles.

    The benefit of the local 3D flux is to provide a sample to characterize the magnitude of the 3D effect since we always do 1D radiative transfer calculations.  The manuscript shows that 1D radiative transfer calculations are performed for all valid retrieved data and as shown in Figure 8 the 3D effect can be substantial under certain conditions.  While it is not a perfect estimate of the radiative fluxes to compare against fluxes derived from BBR, using 3D calculation does seem to be important given the errors when using 1D calculations.

    Existing output from ACM-RT could be used to compute CRE from 1D and 3D radiative transfer calculations.  To demonstrate this we repeated calculations using the Monte Carlo codes in ACM-RT configured to run in 3D and ICA mode and focus on solar CRE since it has larger 3D effects, as shown in the manuscript.

[Figure]

Figure R1. Assessment domain (5x21 km) mean differences for the Hawaii scene between 3D and ICA upward solar fluxes at 20 km for clear-sky conditions computed using Monte Carlo radiative transfer model.

Figure R1 shows the difference in clear-sky solar fluxes which are generally much smaller than differences for cloudy assessment domains (Figure 8 in manuscript). Differences in Figure R1 arise from the 3D calculations seeing surface albedos and atmosphere, i.e., aerosols, outside of the assessment domain that would not be seen by ICA calculations. When using 1D clear-sky fluxes there is will also be some differences due to using a 2-stream solution rather than a Monte Carlo but these should be small.

Figure R2 shows that differences in the clear-sky fluxes are frequently much smaller than the 3D effect on solar CRE, especially for cloudy scenes with larger cloud water paths. This suggests that it should be possible to use clear-sky radiative fluxes computed using the 1D radiative transfer code could be used in combination with the all-sky 3D fluxes to compute the 3D CRE.

[Figure]

Figure R2. Assessment domain (5x21 km) mean differences for Hawaii scene, upper plot is the solar CRE computed using 3D radiative transfer as a function of domain mean cloud water path, while the bottom plot is the difference (3D-ICA) in CRE.

5.  *Why the size of "assessment domain" was chosen as 5´21 km?*

    The size of the domain for the radiative closure is ~100 km$^2$ which was driven in part by performance of the BBR which had its nominal performance requirements defined for this sized domain.  Instead of using a 10 km by 10 km domain, we decided to use a domain that is larger orbit and smaller across orbit.  This is a balance between a domain that is far from the track of the active sensors, for which the scene construction algorithm would have increasing impact on the results, and a narrow but long domain that would make interpretation of the closure difficult.  The use of 21 km along orbit rather than 20 km is due to the 7 km period of the horizontal grid used for for retrievals (Eisenger et al, 2023).

    We have added text and references to the manuscript to explain this choice at the beginning of the Results section

    Eisinger, M., T. Wehr, T., Kubota, D., Bernaerts, and K. Wallace, 2023: The EarthCARE production model and auxiliary products. *Atmospheric Measurement Techniques*, to be submitted.

6.  *The radiative flux distribution will be different by the reference height, which was set at 20 km in the manuscript. Many of readers should not be aware of the importance of the*

*reference height. Can the authors add some explanation of the reason for their choice of the reference height?*

In the operational processing of EarthCARE data, the reference height will be determined in the BMA-FLX processor and certainly be different for each 5x21 km domain.  In this manuscript, which is highlighting the radiative transfer calculations we instead use for simplicity a constant height of 20 km rather than a varying reference height.  A value of 20 km is used in this manuscript because it is shown in Loeb, Kato and Weilicki, 2002, to be an appropriate reference level for flux calculations in Earth radiation budget studies.  Text indicating this is added has been at the beginning of the Results section.

Loeb, N. G., Kato, S., & Wielicki, B. A. (2002). Defining Top-of-the-Atmosphere Flux Reference Level for Earth Radiation Budget Studies, *Journal of Climate, 15*(22), 3301-3309.

7. *Results are probably obtained from synthetic multi-sensor measurements made from GEM. This point should be clarified. In Fig. 3, retrievals are significantly different from the reference (GEM) profile. There are several possible reasons for the discrepancies in this type of experiment. The synthesis measurements are probably superimposed by measurement noise, inversion algorithm should be built on several assumptions and prior information, and the simulation may not be perfect. Please explain the reason of deviation from the reference.*

We believe the referee is asking for an explanation of the differences between the cloud property retrievals and the original GEM data.  This is best explained in the relevant manuscripts documenting the retrieval algorithms.  They are not modified in ACM-COM, they are only prepared for use in the radiative transfer calculations, e.g., ACM-CAP processor output, or combined, e.g., the "composite" profiles created from the single instrument retrievals.

**Specific or typographic comments/questions**

*Eq. (8): Is this the (modified) Gamma distribution? Is there a reference?*

A reference for Equation 8 has been added to the manuscript,

Chylek, P., P. Damiano, and E. P. Shettle, 1992: Infrared emittance of water clouds, J. Atmos. Sci., 49, 1459–1472.

*Fig. 5: SW and LW CREs are usually radiative flux anomaly due to the presence of cloud in the net downward flux at the TOA, and SW and LW CREs are negative and positive, respectively. The authors seem to use unusual definition of the SW and LW CREs. Please give the definition specifically.*

We thank the referee for catching this error creating the plot in the figure. It indeed is reversed from the CREs computed using the standard definition. Figure 5 and associated text has been corrected and updated.

*There are many uses of a word "get", which may be rewritten with more specific word (e.g., "become" and "obtain").*

The text has been modified to reduce the use of the verb "get".

*L419: "W/m2" could be "W $m^{-2}$"*

Fixed.

---

## Referee Report (RR1)

**Review**

This manuscript describes the theoretical foundations of EarthCARE's radiative flux and heating rates product, ACM-RT, that derives from applying a radiative transfer model to aerosol and cloud profiles retrieved from the cloud profiling radar, lidar, and multi-spectral imager. The primary focus is to document the details of the radiative transfer calculations with an emphasis on establishing the value of the unique use of 3D radiative transfer modeling. The subject is appropriate for *Atmospheric Measurement Techniques*, the methods are thoroughly described, and the results are compelling. My only concerns are that (a) in places the paper reads too much like a technical report or theoretical basis document and (b) the paper assumes too much 'insider knowledge' of the EarthCARE products and nomenclature to be fully understood by the general reader. I recommend the paper be published in AMT after the following minor revisions to address these concerns.

Specific Comments

1. The paper reads too much like a report or technical document in some places. The abstract refers to the study as a "report" on at least 3 occasions and the compositing process in Section 3 reads very much like a technical document as opposed to a paper. As opposed to strictly describing a recipe, are there any elements of the thought process or motivating physics that could be described? Similarly, the lists of wavenumber ranges for RRTMG's LW and SW bands in Section 4.1.1 seem out of place in a paper, perhaps they could be converted to a table at least to avoid devoting two paragraphs to lists of numbers. Another example concerns the detailed description of the Lorenz-Mie calculations starting on Line 221 that includes the increments used to step through particle radii. I suggest adopting a more narrative approach throughout the paper to improve readability.

2. While the paper is part of a special issue that likely fills in several additional mission details, I believe this paper should largely stand alone. At a minimum all acronyms should be spelled-out but it would be useful to add a few additional details regarding the ACM-CAP, A-ICE, C-CLD, etc. products. I also didn't see clear definitions of "Hawaii frame" and "Halifax frame".

3. Line 158: it seems one or more words is missing after "ACM-COM's …"

4. While the accuracy of the radiative transfer model and, in particular, the 3D Monte Carlo calculations are discussed at length, a broader discussion of the anticipated sources of error in the ACM-RT product itself owing to retrieval uncertainties and errors in the supplemental meteorological variables and surface albedo is lacking. To what extent do these uncertainties offset the value of modeling 3D effects? I realize the point of the closure studies after launch is to answer this very question, but have any sensitivity studies been conducted to assess the relative magnitudes of geophysical parameter errors vs. radiative transfer errors?

5. What does the black rectangle in Figure 7 represent? It is not described in the caption or in the narrative.

---

## Author Response (AR2)

Response to Referee #3

We thank the referee for their review. Below are the original comments in *italics* with our responses in normal text.

*This manuscript describes the theoretical foundations of EarthCARE's radiative flux and heating rates product, ACM-RT, that derives from applying a radiative transfer model to aerosol and cloud profiles retrieved from the cloud profiling radar, lidar, and multi-spectral imager. The primary focus is to document the details of the radiative transfer calculations with an emphasis on establishing the value of the unique use of 3D radiative transfer modeling. The subject is appropriate for Atmospheric Measurement Techniques, the methods are thoroughly described, and the results are compelling. My only concerns are that (a) in places the paper reads too much like a technical report or theoretical basis document and (b) the paper assumes too much 'insider knowledge' of the EarthCARE products and nomenclature to be fully understood by the general reader. I recommend the paper be published in AMT after the following minor revisions to address these concerns.*

*1. The paper reads too much like a report or technical document in some places. The abstract refers to the study as a "report" on at least 3 occasions and the compositing process in Section 3 reads very much like a technical document as opposed to a paper. As opposed to strictly describing a recipe, are there any elements of the thought process or motivating physics that could be described? Similarly, the lists of wavenumber ranges for RRTMG's LW and SW bands in Section 4.1.1 seem out of place in a paper, perhaps they could be converted to a table at least to avoid devoting two paragraphs to lists of numbers. Another example concerns the detailed description of the Lorenz-Mie calculations starting on Line 221 that includes the increments used to step through particle radii. I suggest adopting a more narrative approach throughout the paper to improve readability.*

For the specific example related to the RRTMG wavenumber intervals, we have put them in a new table. These intervals are not easily found in the literature; listing them here will be useful to readers for they help define computation of all optics described in the paper. We have not changed text related to Lorenz-Mie calculations. As noted in the manuscript, tabulated values of liquid cloud optics are used in ACM-RT. In our opinion, description of the tables is an important detail for some readers.

It is important to note that the manuscripts in this "special edition" essentially double as regular publications and, overly technical, Algorithm Theoretical Basis Documents (ATBDs). Currently, the manuscript is an attempt to straddle these two types of documents. Rewriting to have "a more narrative" style would detract from the balance we believe we have struck.

*2. While the paper is part of a special issue that likely fills in several additional mission details, I believe this paper should largely stand alone. At a minimum all acronyms should be spelled-out but it would be useful to add a few additional details regarding the ACM-CAP, A-ICE, C-CLD, etc. products. I also didn't see clear definitions of "Hawaii frame" and "Halifax frame".*

Explicit definitions of the "Halifax" and "Hawaii" frames have been added at the beginning of Section 5. We have expanded all acronyms and abbreviations when they first appear. We have selectively modified the text to identify what is being used from EarthCARE products that are not the focus of the manuscript.

*3. Line 158: it seems one or more words is missing after "ACM-COM's …"*

The sentence was adjusted to make it clear that we are discussing "ACM-COM profiles".

*4. While the accuracy of the radiative transfer model and, in particular, the 3D Monte Carlo calculations are discussed at length, a broader discussion of the anticipated sources of error in the ACM-RT product itself owing to retrieval uncertainties and errors in the supplemental meteorological variables and surface albedo is lacking. To what extent do these uncertainties offset the value of modeling 3D effects? I realize the point of the closure studies after launch is to answer this very question, but have any sensitivity studies been conducted to assess the relative magnitudes of geophysical parameter errors vs. radiative transfer errors?*

We agree with the reviewer that propagation of input uncertainties through radiative transfer models is important. We are in the process of preparing a manuscript for the ACMB-DF (radiative closure) processor that will show errors in radiative quantities relative to reference values computed from the test scenes. This will, however, lack assessment of errors in the retrievals, meteorology, and surface properties. The next step will be incorporation of input errors (uncertainties) and their impacts on computed radiative fluxes and radiances. We are simply not prepared right now to present results pertaining to these issues.

Provision of quantitative estimates of radiative uncertainties requires both characterization of inputs errors and a methodology to utilize them. For example, it is not enough to know just the magnitudes of input uncertainties, which are challenging in their own right, but one also needs to consider correlations in uncertainties in both space and time. We expect that the study of propagation of input uncertainties through the radiative transfer models will be sufficiently complex to warrant a dedicated manuscript. That said, as presented in this manuscript, using 3D instead of 1D radiative transfer removes that source of uncertainty from computed radiative fluxes and radiances.

*5. What does the black rectangle in Figure 7 represent? It is not described in the caption or in the narrative.*

The rectangles indicate the size of assessment domains over which radiative transfer results are averaged for the radiative closure. This has been made clear in the figure caption.